# *Porphyromonas gingivalis* Bundled Fimbriae Interact with Outer Membrane Vesicles, Commensals and Fibroblasts

**DOI:** 10.3390/ijms27010383

**Published:** 2025-12-30

**Authors:** Julian Lambertz, Eva Miriam Buhl, Christian Apel, Christian Preisinger, Georg Conrads

**Affiliations:** 1Division of Oral Microbiology and Immunology, Department of Operative Dentistry, Periodontology and Preventive Dentistry, Rheinisch-Westfälische Technische Hochschule (RWTH) University Hospital, 52074 Aachen, Germany; julian.lambertz@rwth-aachen.de; 2Electron Microscopy Facility, Institute of Pathology, Rheinisch-Westfälische Technische Hochschule University Hospital, 52074 Aachen, Germany; ebuhl@ukaachen.de; 3Department of Biohybrid & Medical Textiles (BioTex), Institute of Applied Medical Engineering, Rheinisch-Westfälische Technische Hochschule (RWTH) University Hospital, 52074 Aachen, Germany; apel@ame.rwth-aachen.de; 4Proteomics Facility, Interdisciplinary Centre for Clinical Research (IZKF), RWTH Aachen University Hospital, 52074 Aachen, Germany; cpreisinger@ukaachen.de

**Keywords:** *Porphyromonas gingivalis*, bundled fimbriae, FimA, FimCDE, outer membrane vesicles (OMVs), fimbriae-associated OMVs (FAVs), outer membrane extensions (OMEs)

## Abstract

*Porphyromonas gingivalis* plays a key role in periodontal disease and has been associated with several serious systemic diseases. Its fimbriae are a major virulence factor. We recently demonstrated the formation of bundles of long FimA fimbriae in strain ATCC 33277. Transmission (TEM) and scanning electron microscopy (SEM) were used to examine a collection of *P. gingivalis* strains representing all seven known FimA types (I, Ib, IIa, IIb, III–V) and a *P. gulae* strain (type A). Additionally, two *P. gingivalis* strains (ATCC 49417 and OMI 1127) were investigated in dual-species approaches together with *Fusobacterium nucleatum* or *Streptococcus oralis* as well as in co-culture with human gingival fibroblasts (HGFs). To evaluate the role of fimbriae accessory proteins FimCDE, proteomic analysis of outer membrane vesicles (OMVs) was performed. Bundling was confirmed to occur regardless of FimA type but was impaired by strong capsule formation. Furthermore, tubular and chain-like outer membrane extensions (OMEs) were identified in most strains examined, including *P. gulae*. For the first time, fimbriae-associated OMVs (FAVs) were observed. REM images suggest that bundled fimbriae, OMEs and FAVs form connections with *F. nucleatum* and *S. oralis*. Proteome analysis of OMV content revealed the ratios of FimA to accessory proteins to be approximately 13:1 for FimC and FimD and approximately 7:1 for FimE. The results imply more accessory proteins per fimbriae or shorter FimA fimbriae in OMVs than in cells. Since FimCDE are known to be responsible for the adhesion properties and autoaggregation of FimA fimbriae, we propose that they could also mediate the stability of bundled fimbriae and the binding of OMVs.

## 1. Introduction

*Porphyromonas gingivalis* (*P. gingivalis*) is of great significance in the pathogenesis of periodontal diseases (PDs), a group of polymicrobial inflammatory diseases that lead to gradual degradation of the periodontal ligament [1]. In addition, PDs and *P. gingivalis* have been linked to conditions outside the oral cavity, including cardiovascular disease [2], rheumatoid arthritis [3], neurodegenerative diseases [4,5], chronic kidney diseases [6] and adverse pregnancy outcomes [7].

*P. gingivalis* has been identified as a keystone pathogen due to its ability to cause a shift in the composition of the oral microbiome, even in low abundance, transforming it from a healthy state to one characterized by dysbiosis [8,9]. It decouples the inflammatory response, which is vital for the provision of nutrients, from the bacteriotoxic (killing) immune reaction. For instance, *P. gingivalis* has been observed to suppress phagocytosis whilst concomitantly leading to the upregulation of inflammatory cytokines, thereby promoting the progression of periodontitis whilst simultaneously ensuring the survival of periodontal pathogens [10,11,12,13,14].

*P. gingivalis* fimbriae are among the most potent pathogenic factors for host colonization [15] and evasion of the innate immune system [16]. *P. gingivalis* expresses various types of fimbriae by different gene clusters: first, the FimA fimbriae, also termed “major” or “long fimbriae”, and second, the Mfa1 fimbriae, also termed “minor” or “short fimbriae” [17,18]. The nomenclature derives from their stalk monomer protein, FimA or Mfa1, respectively. A third adhesin, identified as PGN_1808 in ATCC 33277 and PG1881 in W83, also known as Ffp1 (filament-forming protein), was recently characterized and found to form polymeric fimbriae in an overexpressing mutant [19,20,21]. All three types have been classified as type V pili, which appear to be unique to the Bacteroidia class [22]. A recent structural analysis of polymerized pili has provided evidence that type V pili are assembled by a protease-mediated strand exchange mechanism at the cell surface [21,23].

Long fimbriae consist of anchor (FimB), stalk (FimA) and accessory (tip) proteins (FimCDE). The role of FimA has been investigated in depth [24,25,26]. The FimA fimbriae have further been classified into six genotypes (I-V, Ib) [27,28,29]. By introducing a novel typing method, our group recently identified the necessity for genotype II to be further divided into two subtypes (IIa and IIb) owing to disparities in the amino acid sequence within functional domains [25]. Sequence variations in the FimA stalk protein exert an effect on the tertiary structure of the fimbrial protein. For instance, loop regions of types I, II and IV exhibit significant structural variations [23].

FimB has been demonstrated to regulate the length and expression of the fimbriae and to anchor them to the cell surface [23,30]. This also leads to an immunostimulatory effect that is absent in strain ATCC 33277 (FimB^−^) but present in strain 381 (FimB^+^) [31]. In contrast, the exact role of the accessory proteins (FimC, D and E) remains to be elucidated. It has been established that they are non-essential for the polymerization process because long fimbriae are formed in knock-out mutants [32,33]. In more detail, a *fimE* insertion mutant still produced FimC and FimD, but neither protein was mounted on the lengthening fimbriae. This finding suggests that FimE is required for the assembly of FimC and FimD onto the pilus. Furthermore, *fimC*, *fimD* and *fimE* insertion mutants exhibited a loss of autoaggregation abilities. Purified fimbriae from all three mutants had diminished binding activities to glyceraldehyde-3-phosphate dehydrogenase (GAPDH) from *Streptococcus oralis* (*S. oralis*), fibronectin and type I collagen [33]. Another study demonstrated that fimbriae from the isogenic mutant lacking FimCDE showed a markedly diminished capacity to inhibit IL-12p70 production, invade host cells and persist intracellularly. This reduction was attributed to the mutant’s inability to interact with complement receptor 3 (CR3)—though its interaction with TLR2 remained unaffected—and it also triggered significantly less alveolar bone loss than the wild type [34]. Furthermore, the mutant failed to exploit CXC-chemokine receptor 4 (CXCR4) to suppress TLR2 signaling, resulting in impaired immune subversion in vivo. Purified FimC and FimD (but not FimE) were shown to interact with CXCR4 [35]. The FimCDE complex is modified by *P. gingivalis’* peptidyl arginine deiminases (PPADs), which is essential for proinflammatory TLR2 activation [36]. Furthermore, FimCDE, but not FimA, were shown to be responsible for the invasive capabilities of *P. gingivalis* OMVs [37].

Together, these reports indicate the importance of FimCDE for the assembly of fully functional fimbriae and ultimately in the virulence of *P. gingivalis*. The tertiary structure of functional Fim domains, in turn, might exert an influence on the structure of matured fimbriae and their interactions with the host and other oral bacteria. The study of Meyer et al. [25] revealed a wide spectrum of fimbrial superstructures for the first time, including bundling, cell–cell knotting (*P. gingivalis* cells connected via bundled fimbriae) and brick-wall formation (regular, brickwork-like arrangements of cells) in strain ATCC 33277 [25]. The bundling and cell–cell knotting of long fimbriae were long known to represent a virulence mechanism, utilized by EPEC/EHEC and *Salmonella* as prominent examples. These mechanisms are instrumental in facilitating surface adhesion and coaggregation [38,39,40].

It was our intention to visualize the formation of bundled fimbriae in different *P. gingivalis* FimA types and to monitor interactions with OMVs as well as with commensal bacteria such as *S. oralis* and *Fusobacterium nucleatum* (*F. nucleatum*) and with human gingival fibroblasts (HGFs). Furthermore, the protein content of OMVs from two strains (ATCC 49417 and 84Pg1) was analyzed, with a special focus on fimbrial proteins including the under-investigated FimCDE. Our hypothesis here is that FimCDE may play a mediating role in the adhesion of OMVs to bundled FimA fimbriae.

## 2. Results

### 2.1. Bundling of FimA Fimbriae

To determine whether strains other than ATCC 33277 are capable of forming bundled fimbriae, eight strains of *P. gingivalis* plus one representative of *P. gulae* from our strain collection (Table 1) were cultivated in Brucella broth in microtiter plates and examined using scanning electron microscopy (SEM) and transmission electron microscopy (TEM). The selection includes representatives of all seven *fim* types of *P. gingivalis* (types I, Ib, IIa, IIb, III, IV, V) and type A of *P. gulae*. The latter shares a close relationship with the type Ib cluster of *P. gingivalis* strains, and it is hypothesized that it is potentially its ancestral form [25,41].

The bundling of long fimbriae was confirmed in ATCC 33277 (OMI 1132, type I) but was also clearly identifiable in OMI 1127 (84Pg1, Ib), OMI 883 (IIa), OMI 778 (IIb), ATCC 49417 (OMI 1071, III) and OMI 622 (V), representing FimA types I, Ib, IIa, IIb, III and V (Figure 1). However, in the representative of type IV, OMI 1049 (Figure 2A,D), the cell connections were found less frequently and appeared more amorphous, though they could still be assumed to be bundled fimbriae given their elongated, narrow, hourglass-like shape (Figure 2A). Compared to the others, this strain’s cultures appeared slimy on agar, suggesting increased production of capsule polysaccharides, and such slime could explain the amorphous appearance. The bundling of long fimbriae was not visualized in W83 (OMI 629, type IV) (Figure 2B,E) or OMI 1080 (*P. gulae*, fimA type A) (Figure 2C,F). Strain W83 is known to be afimbriated, very likely due to a mutated FimS histidine kinase [42].

Taken together, the bundling of long fimbriae occurs most likely in all *P. gingivalis* strains that express FimA fimbriae regardless of the type. A strong capsule reduces the probability of bundling, as seen in the type IV strain. The micrographs, however, did not provide evidence for bundling of type A fimbriae in *P. gingivalis’* ancestor *P. gulae*.

### 2.2. Tubular and Chain-like Outer Membrane Extensions as well as Fimbriae-Associated OMVs

A modified method enabled a more accurate representation of the OMVs and fimbriae. At the start of incubation, the TEM grids were placed in the wells of the microtiter plates so that the bacteria could grow directly on them. This reduced artifactual mechanical damage to the fine structures during preparation for the electron microscopic examination. With this modification, electron micrographs revealed diverse morphologies of OMVs, deviating from the typical spherical shape.

Firstly, long tubular extensions of the outer membrane (OMEs) were observed protruding from the bacterial surface. These tubes exhibited either kinked (Figure 3A), winding (Figure 3C) or straight (Figure 3E) conformations. A prominent electron-dense surface layer (EDSL: an outer membrane-located, immunostimulant, granular protein-rich layer, which constitutes a major portion of *P. gingivalis* OMV’s volume [43,44]) was apparent on the tubular extensions and is most clearly visible in Figure 3B,D,E.

Secondly, chains of OMVs were found adhering to the bacterial surface (e.g., Figure 3B,D,G,I), with the individual vesicles appearing to be fused or in close continuity. These chains were also observed in cell-detached form (e.g., Figure 3B,F), but this could have happened during preparation. There were transitional forms suggestive of tubular-to-vesicle (and/or vesicle-to-tubular) transformation (e.g., Figure 3H,I). Interestingly, the tubular and chain-like structures often featured vesicles of larger diameter at their distal ends (labeled with ▷ in Figure 3), indicative of a terminal flow of vesicles and its load. As shown in Figure 3J, tubes were observed to be attached to bundles of fimbriae. Tubular and chain-like OMV structures were found in *P. gingivalis* strains ATCC 33277 (type I), OMI 1127 (Ib), OMI 883 (IIa), ATCC 49417 (III), W83 (IV), OMI 1049 (IV) and OMI 622 (V) and in the *P. gulae* strain OMI 1080 (A).

Furthermore, spherical OMVs were observed attached to individual or bundled fimbriae. They were located both at the end (e.g., Figure 4I) and in the course (e.g., Figure 4G) of the long (bundled) fimbriae. This phenomenon, hereinafter referred to as “fimbriae-associated OMVs” (FAVs), were detected in *P. gingivalis* strains ATCC 33277 (I), OMI 1127 (Ib), OMI 883 (IIa), ATCC 49417 (III) and OMI 622 (V).

Taken together, tubular OMEs, OMV chains and FAVs were found across diverse *P. gingivalis* strains and, with the exception of the FAVs, also in *P. gulae*. Importantly, OMV–fimbriae interactions do not appear to depend on the FimA type, as the phenomenon was observed in strains expressing types I, Ib, IIa, III and V, with types Ib and V being phylogenetically the most divergent [25]. The masking and the less frequent abundance of this phenomenon in the examined type IV strain is likely attributable to the robust capsule and the associated diminished number of fimbriae, as previously described.

### 2.3. Interactions with S. oralis and F. nucleatum

To assess whether the observed fimbrial superstructures also occur between *P. gingivalis* and oral commensals, *P. gingivalis* was co-incubated with either *S. oralis* or *F. nucleatum* as model organisms of the oral biofilm. *S. oralis* has been identified as an early colonizer of dental plaque [45], and FimA fimbriae have been reported to play a crucial role in coaggregation with *S. oralis*, primarily via *S. oralis*’ GAPDH [46,47]. *F. nucleatum,* on the other hand, has been referred to as a “bridge” organism given its ability to co-aggregate with early and late colonizers in dental biofilms [48].

Figure 5 illustrates the SEM micrographs of both co-cultures. To facilitate identification, *P. gingivalis* was colored purple in this image. The co-culture of *S. oralis* with *P. gingivalis* was interspersed with *P. gingivalis* nests and individual cells. Clear contacts with *S. oralis* could be identified. It was observed that some of these appeared to be based on fimbriae, as demonstrated in Figure 5A,C (marked with “F” in the figure). Others appeared to consist of the OMV chains described in the previous paragraph, as illustrated in Figure 5A (denoted with “C”). However, it is important to note that these structures could also be bundled fimbriae with attached OMVs. It was not possible to make a clear assignment because, unlike in the pure *P. gingivalis* cultures, the (bundled) fimbriae were masked by an extracellular matrix and/or OMVs (e.g., Figure 5B).

The *F. nucleatum* co-culture was also interspersed with *P. gingivalis*. Few (bundled) fimbriae were observed. Interestingly, some *P. gingivalis* showed cell-morphological changes when attached to fusobacteria cells. The bacterium was asymmetrical, with one part elongated and tubular—possibly attached to the fusobacterial cell by a foot-like bundle of fimbriae—and the other cap-shaped (Figure 5E,F, marked with “C/F” in G).

Thus, *P. gingivalis* was able to interact with both *S. oralis* and *F. nucleatum*. The interactions seemed to consist of bundled fimbriae. Chains of OMVs and/or bundled fimbriae with associated OMVs were also observed to connect *P. gingivalis* with both commensal species.

### 2.4. Interactions with Gingival Fibroblasts

The long fimbriae of *P. gingivalis* have been identified as a specific TLR2 target in primary human gingival fibroblasts. In particular, the tip protein complex FimCDE has been previously demonstrated to be indispensable for TLR2 activation [36,49]. Despite the challenges of co-culturing anaerobic bacteria and aerobic cells, an effort was made to visualize the described interactions by employing various culture media and growth conditions. To this end, HGFs were co-incubated anaerobically with each of the two different *P. gingivalis* strains (ATCC 49417 and OMI 1127) for 24 h. The cultures were again examined using SEM. Compared to the pure fibroblast control culture (Figure 6B), the fibroblasts showed stress and rounded up during co-incubation with ATCC 49417 (Figure 6A), most likely because of high bacterial cell numbers. Interestingly, no such stress reaction occurred during co-incubation with OMI 1127, growing slower and in lower cell numbers. Here, the fibroblasts exhibited a flattened, dendritic configuration with multiple cytoplasmic extensions as indicator for more regular growth.

The images revealed strong colonization of the fibroblasts by ATCC 49417, which accumulated/attached on the fibroblast surface (Figure 6C), indicating fimbriae-mediated adhesion. Thick filamentous connections between ATCC 49417 and HGFs are visible in the images, most clearly in Figure 6E, which is an enlargement of Figure 6D. While the braid-like structures evident in the TEM images in Figure 1 and Figure 4 are not visible at this magnification, the thickness clearly indicates that these are bundled fimbriae. Additionally, direct interactions between fibroblast extensions (dendritic extensions and thin filopodia) and bacterial cells were visible (Figure 6G). Interestingly, similar to the co-cultures of *P. gingivalis* with *F. nucleatum*, stalked adhesion with partially asymmetrical reformation of cells (Figure 6F) was observed, especially in ATCC 49417. Strikingly, OMI 1127-HGF and ATCC 49417-HGF interactions differed. OMI 1127 cells were primarily found in protein-rich ECM secretions (e.g., collagen, elastin and fibronectin) in the preparation (see Figure 6H,I).

Taken together, the SEM micrographs illustrate the situation and interactions, including bundled fimbriae, during the co-incubation of *P. gingivalis* with HGFs under anaerobic conditions for the first time, offering a novel perspective.

### 2.5. Quantities and Ratios of FimA and FimCDE in OMVs

The levels of FimA, FimC, FimD and FimE in the OMVs of two *P. gingivalis* strains, ATCC 49417 and OMI 1127, were determined. For this purpose, the OMVs were isolated from the culture medium by filtration and differential centrifugation. The pellet obtained was subjected to proteome analysis.

In order to estimate absolute protein quantities, the intensity-based absolute quantification (iBAQ) algorithm (see [50], Supplementary Methods) was used. The iBAQ values were determined by analysis of the mass spectrometry raw data with Spectronaut, and the calculated mean values (MVs) are displayed in Appendix A. Figure 7A,B shows the data graphically. The iBAQ values were also used to analyze the ratios of the shaft protein to each accessory protein. The mean ratios (MRs) are shown in Figure 7C,D. The iBAQ MVs in ATCC 49417 OMVs were, by an approximate factor of ten, greater than those in OMI 1127 (Appendix A), but this was due to technical reasons, and the quantities should not be interpreted as absolute values. Conversely, the MRs demonstrate only minor disparities between the same proteins from the two strains. The MRs of FimA to FimC and FimD were similar in both strains. For FimA to FimC, they amount to 10.65 (SD: 0.72) and 16.09 (SD: 1.79), respectively, and for FimA to FimD, 11.83 (SD: 0.77) and 14.16 (SD: 2.09), respectively. The differences between the proportions of FimC and FimD were not statistically significant. However, the MRs of FimA to FimE were found to be significantly lower than those of FimA to FimC (both *p* < 0.01) and FimD (*p* < 0.01, *p* < 0.05) in both strains. They were 7.16 (SD: 0.64) and 6.29 (SD: 0.71), respectively. Thus, FimE was observed to be more prevalent than FimC and Fim D in the OMVs of both strains.

In summary, the data suggests that while, for technical reasons, the amounts of fimbrial proteins FimA, C, D and E vary in OMVs of different *P. gingivalis* strains and preparations, the ratios between the shaft protein to the accessory proteins remain constant. In conclusion, OMVs contain more FimE than FimC and FimD, while FimC and FimD occur in equal proportions. The ratio of FimA to FimE was approximately 7:1 and for FimA to FimC and FimD approximately 13:1.

## 3. Discussion

In this study, we generated a comprehensive set of SEM and TEM micrographs of *P. gingivalis*, including both mono- and dual-species cultures with oral commensals and HGFs. These images provided structural insights into the organization of fimbriae, the diverse morphology of OMVs and the interactions between these two important virulence factors. Figure 8 summarizes the observed structures. In parallel, analyzing the content of *P. gingivalis* OMVs allowed us to draw new conclusions regarding the role and spatial arrangement of the accessory fimbrial proteins FimC, FimD and FimE. These findings further elucidate their potential contributions to fimbrial architecture and to host or interspecies interactions.

### 3.1. Confirmation of the Bundling Phenomenon Across a Diverse Range of FimA Types

The bundling of long FimA fimbriae of *P. gingivalis* was recently discovered only for strain ATCC 33277 by Meyer et al. [25]. The findings here reveal that this phenomenon is a common feature among *P. gingivalis* strains, regardless of *fimA*/FimA geno-/serotype. Thus, the differences in the tertiary structure among different types do not impede the bundling. The micrographs provided here unambiguously show the bundling of the thin filaments to braid-like structures, particularly impressive in Figure 1B,C,E,F and Figure 4C,F. In principle, one could discuss whether the bundled fimbriae were formed by long FimA fimbriae only (what we conclude) or if short Mfa1 fimbriae and/or Ffp1 might also be involved. However, since short Mfa1 fimbriae could only be visualized by electron microscopy in *fimA*-deficient mutant strains [18] and Ffp1 fimbriae in overexpressing mutant strains [19], it can be assumed with a high degree of certainty that the bundled fimbriae consist of mainly long FimA fimbriae.

The fact that bundles of fimbriae were reliably identifiable in all types with the exclusion of type IV and *P. gulae* seems to be due to the mucilaginous capsule overlay of the latter two. The influence of the capsule on the fimbriae can also be deduced from the literature. Firstly, it has been shown that non-encapsulated *P. gingivalis* strains have stronger adhesive properties than encapsulated strains [51]. In another study, capsule production inhibited biofilm formation on polystyrene surfaces, as demonstrated by a glycosyltransferase mutant that lost its capsule but showed increased biofilm formation [52]. Thus, the presence of a capsule reduces the adhesive properties, which are known to be mediated by fimbriae, and thus leads to less biofilm formation. This was also confirmed in the experiments conducted here. Compared to all other fimbriated strains, the bacterial cell count was significantly reduced in encapsulated strains W83, OMI 1049 and the *P. gulae* strain. However, the following questions cannot be answered by our data: (i) Do the encapsulation properties merely “hide” regular fimbriae? (ii) Is the fimbriae polymerization restricted by capsular and “slime” production? (iii) Is FimA less or not expressed in these strains? Conversely, if FimA is not expressed, as in W83, this could possibly stimulate capsule production. The literature does not provide a general answer, but several studies with other bacteria (e.g., *Klebsiella pneumoniae*, *Streptococcus pneumoniae*) have suggested that capsules physically impair the pili/fimbriae [53,54]. However, reciprocal regulatory mechanisms, as known from other bacterial species, should not be neglected [55]. In addition to a very likely physical interference between both virulence factors, the expression of fimbriae and capsules in *P. gingivalis* may also be dynamically regulated and depend on the bacteria’s immediate environment. Further studies are required to investigate the mutual influence between capsule and fimbriae formation in *P. gingivalis*, especially with regard to expression regulation.

It was particularly difficult to visualize the bundled fimbriae in contact with *S. oralis*, *F. nucleatum* (Figure 5) and HGFs (Figure 6) since cells of different species appeared to be able to bind much more closely than two *P. gingivalis* cells (Figure 1). Nevertheless, our images indicate that, as expected, these fimbriae-mediated connections are indeed formed with oral commensals and host cells.

### 3.2. New Insights in OMV Morphologies/OMEs

It has long been known that *P. gingivalis* forms OMVs and releases them into the environment [56,57,58]. Nevertheless, the underlying mechanism is not yet fully understood [59,60]. In this study, OMV morphologies that go beyond the frequently described blebbing were observed. The electron micrographs revealed tubular and chain-like forms of OMVs in almost all investigated strains, including *P. gulae*. Such membrane extensions have previously been described for other bacteria. Kaplan et al. [61] examined electron cryo-tomograms of approximately 90 Gram-negative bacterial species and found extensions of the outer membrane in 13 of them. The authors subdivided these into seven forms, some of which were found to occur concomitantly within the same bacterial species [61]. According to them, the presumed functions are also very diverse [61]. Fischer et al. [62] observed tubular extensions protruding from the cell surface in a *Flavobacterium* strain phylogenetically related to *P. gingivalis*, which transformed into chains of interconnected vesicles during the stationary growth phase by a process known as biopearling [62]. The authors assumed that the outer membrane appendages serve to expand the effective cell surface and thus also increase the number of virulence factors associated with the outer membrane. This could favor both nutrient uptake and the virulence of the bacterium, as a high surface-to-volume ratio is favorable for nutrient supply [62,63]. Furthermore, the appendages would lead to a reduction in the diffusion distance between membrane proteins and molecules or surfaces and increase the immediate surrounding volume, potentially facilitating substrate binding and processing [62].

In addition, we observed chains of OMVs with one larger vesicle at the distal end (e.g., Figure 3F,H,I). Assuming that the lumina are connected, this observation may be indicative of a “flow” of contents in a distal direction. Together with the fact that tubes and chains of OMVs were identifiable as connections between *P. gingivalis* and *S. oralis* or *F. nucleatum*, respectively (see Figure 5A,B,E), one could suggest that the chains and tubes, especially in multispecies biofilms such as those found in the gingival pocket, could serve to direct the distribution of virulence factors (such as highly proteolytic and toxic gingipains) to other bacteria or host cells.

As outlined, extensions of the outer membrane (OMEs) were observed protruding from the bacterial surface. For a detailed structural elucidation of the OMEs, electron cryo-tomography (ECT) should be used in further investigations. This technique produces higher-resolution images than conventional TEM and is therefore also recommended by Kaplan et al. for further investigation of these structures [61]. For future studies, it remains to be clarified to what extent all these findings are transferable to *P. gingivalis* OMV morphologies. Furthermore, it needs to be determined which mechanisms, genes and proteins lead to this phenomenon and by which structures the OMVs remain connected as a chain. It also remains unclear whether the tubes are an intermediate in the pearling process or a stable form. Clarification of this circumstance could also lead to conclusions about the function. Indeed, biopearling was recently demonstrated by Vermilyea et al. for *P. gingivalis* strain 381 [64]. Our investigation of phylogenetically distant *P. gingivalis* strains thus provides further evidence that biopearling is a widespread phenomenon in *P. gingivalis*.

Finally, at the systemic level, *P. gingivalis* gingipain-rich OMVs increase vascular permeability, which may explain their involvement in systemic diseases [65]. Furthermore, they have been shown to disrupt tight junctions, impairing the barrier function of lung epithelial cells and the blood–brain barrier [66,67]. In vivo, OMVs translocated to the brain and induced Alzheimer’s disease-like neuropathological changes through gingipain-driven neuroinflammation [68]. These systemic effects are another reason why more research on OMV content is urgently needed.

### 3.3. FAVs and the Role of FimCDE

The electron micrographs also showed that *P. gingivalis* OMVs were frequently associated with long fimbriae and appeared to be bound to them. These FAVs were observed in almost all *P. gingivalis* strains examined but not in the *P. gulae* strain.

To the best of our knowledge, there is no study so far that has reported direct contact between OMVs and FimA fimbriae. Nevertheless, the FAVs observed here seem to represent a new and plausible biological mechanism, and we suggest that FimCDE could play a crucial role here. It is known that FimCDE is essential for the adhesive properties of long fimbriae and responsible for the autoaggregation abilities of FimA fimbriae, not the FimA itself, since long fimbriae from mutants without FimCDE lose their autoaggregation abilities [33]. Moreover, FimCDE are present in OMVs [37] (our own data presented in Section 2.5). Thus, it appears reasonable to hypothesize that FimCDE could play a mediating role in the adhesion of OMVs to FimA fimbriae. Furthermore, the observation that OMV–fimbriae interactions occurred irrespective of FimA type in phylogenetically distant strains provides additional support for this concept since *fimCDE* appears to be less divergent than *fimA*, with only two known genotypes, except for point mutations [69].

Our hypothesis is that autoaggregation occurs between FimCDE complexes of OMVs and long bundled fimbriae. Until now, it has been assumed that FimCDE complexes were located at the tip of the long fimbriae. In this study, we observed that OMVs also attached in the course of bundled fimbriae. One potential explanation for this phenomenon is that the twisting and braid formation of fimbriae of different lengths or distant anchors in the membrane result in the FimCDE complexes of each fimbria being distributed over a range of locations in the bundles (Figure 9A). In the proteome analyses, the ratios of FimA to the three accessory proteins in OMVs from two *P. gingivalis* strains were determined. The results suggest that the tip complex follows after approximately 13 polymerized FimA molecules (based on the ratio of FimA to FimC and FimD, Figure 7). A key observation in our study was the increased ratio of FimE compared to FimC and FimD in the OMVs of two examined *P. gingivalis* strains. Little is known about the specific functions of the individual accessory proteins. There are indications that FimE is responsible for the assembly of the tip complex rather than for the adhesive properties to host molecules [33,35]. However, our findings imply that (a subfraction of) “long” fimbriae present on the OMVs must be considerably shorter than those on the bacterial cell surface. Assuming an average size of 4 nanometers for a FimA molecule [17], the OMV–fimbriae shaft consisting of 13 polymerized FimA molecules measures approximately 50 nanometers in length.

In summary, we propose the following theory. A fimbriae bundle contains FimCDE tip proteins at different positions, possibly a consequence of the twisting of multiple fimbriae (Figure 9A). OMVs possess shorter “long” fimbriae on their surface. Complexes are formed between FimCDE of the long *P. gingivalis* (parent cell) fimbriae and FimCDE of the short OMVs’ FimA fimbriae (Figure 9B). Furthermore, these fimbriae-spanning complexes may also be responsible for the formation and stability of the bundled fimbriae (Figure 9C). An alternative explanation for the occurrence of OMVs in the course of fimbrial bundling and, at least to some extent, for the altered, reduced ratios of FimA to the accessory proteins could be that FimCDE complexes are not only located at the tip of the fimbriae, as it has been postulated so far [26,33], but also down- (anchor-) stream. While the localization of the Mfa3 protein as part of the short fimbrial tip complex has been verified to be at the tip of the fimbrial shaft [70], there is actually no study that has determined the exact localization of FimCDE. This theory is supported by the study by Nagano et al. [30]; the authors determined the molar ratios FimA:FimC:FimD:FimE to be 20:1:1:1 in whole-cell lysate [30]. Apparently, this relatively low ratio of FimA to FimC-E in contrast to the enormous length of FimA polymers is not explainable by the tip location exclusively. Consequently, the authors concluded that two FimCDE complexes are present in one FimA pilus [30]. To provide substantiating evidence, the development and utilization of fluorescently labeled monoclonal antibodies targeting FimCDE would be necessary to test the hypothesis. Ultimately, a synergy between both phenomena (firstly, different positions where FimCDE complexes come to lie, and secondly, multiple FimCDE complexes along the long fimbriae) would also be plausible.

Additionally, the function of FAVs remains to be elucidated. The images provided herein show evidence that FAVs and OMVs may support the chemical communication between *P. gingivalis* and *S. oralis* or *F. nucleatum*, respectively (e.g., Figure 5A–C,E). In this context, bundled fimbriae may function as guiding lines, directing OMVs towards other bacteria within the biofilm or towards host surfaces. Indeed, FimA fimbriae have an important role in initial biofilm formation [71] and attachment to other bacteria in oral biofilm (e.g., *Actinomyces viscosus*, *Treponema denticola*, *S. oralis* [72,73,74], *F. nucleatum* (present study)) and to host cells [75,76,77]. Moreover, virulence factors are enriched in bacterial OMVs [78]. Gingipain levels were found to be threefold higher in OMVs of *P. gingivalis* than in cell surface extracts [37].

Thus, the directed transport of OMVs along (bundled) FimA fimbriae could be advantageous for a targeted distribution of virulence factors. Subsequent studies are necessary to validate this hypothesis.

## 4. Materials and Methods

### 4.1. Bacterial Strains and Growth Conditions

This study used a selection of human *P. gingivalis* isolates and an isolate of the animal analog *P. gulae* from our strain collection, covering the full spectrum of *fimA* genotypes. To investigate the fimbrial interactions between *P. gingivalis* and other oral bacteria present in the periodontal pocket, one *S. oralis* and one *F. nucleatum* strain were used in a co-incubation experiment. The designations and *fimA* subtypes of the strains investigated are shown in Table 1. From the cryotubes stored at −80 °C, all isolates were cultured at 37 °C anaerobically on tryptone soya blood agar (TSBA, Thermo Fisher Scientific Inc., Waltham, MA, USA) for one week. A few pure colonies were used to inoculate a volume of 200 µL supplemented Brucella broth medium (BBL™ VitK_1_—Hemin Solution, Becton, Dickinson and Company, Franklin Lakes, NJ, USA) in 48-well microplates (Thermo Fisher Scientific Inc.), and the liquid cultures were again incubated anaerobically (<1% oxygen, ≥13% CO_2_ (GasPak™ EZ, Becton, Dickinson and Company) at 37 °C for 5–7 days.

### 4.2. Co-Incubation with Oral Commensal Bacteria

The four strains, ATCC 49417, OMI 1127, OMI 581 (*S. oralis*) and OMI 1040 (*F. nucleatum*), were initially incubated individually on TSBA (all at 37 °C anaerobically), as described in Section 4.1. After inoculation, co-incubation was carried out in liquid culture, as described in Section 4.1. Monocultures were always used as contrast. The samples were examined using SEM according to the method described in Section 4.4.

### 4.3. Co-Incubation with Gingival Fibroblasts

To investigate the occurrence of bundled fimbriae in the presence of typical periodontal host cells, cultures of two *P. gingivalis* strains, ATCC 49417 and OMI 1127, were co-incubated with immortalized HGFs (P10866-IM, Innoprot, Derio, Spain). The bacteria were cultured five days prior to inoculation, as described in Section 4.1. The HGFs (15,000 or 30,000 cells/well) were grown on sterile glass disks (Ø = 15 mm) (Menzel Gläser, Epredia, Braunschweig, Germany) in 24-well microtiter plates (Thermo Fisher Scientific Inc.) and filled to 800 µL with DMEM (GIBCO DMEM + GlutaMAX^TM^-I, Thermo Fisher Scientific Inc.) supplemented with 10% fetal bovine serum (GIBCO Fetal Bovine Serum (FBS), Thermo Fisher Scientific Inc.). No gentamicin was added. Incubation was performed for two days at 37 °C under a supply of 5% CO_2_. Then, the medium of the fibroblast culture was replaced with a mixture of 75% DMEM (with 10% FBS) and 25% Brucella broth medium (supplemented with VitK_1_ and Hemin). Preliminary testing demonstrated that this mixture is the optimal compromise between bacterial and fibroblast growth medium. Two to three bacterial colonies from the agar plates were suspended in Brucella broth medium in Eppendorf tubes and utilized to inoculate the fibroblasts. Plates were incubated anaerobically (which was found to be tolerable for HGFs for a limited time), as described in Section 4.1, for 24 h and then examined by SEM, as described in Section 4.4. All approaches were performed in triplicate.

### 4.4. Electron Microscopic Examination

Bacterial cultures were examined using TEM and SEM as described in our previous study [25]. For TEM examination, the samples in the microplates were rinsed with 0.1 M HEPES buffer and transferred to formvar carbon-coated glow-discharged nickel grids (Maxtaform, 200 mesh; Science Services GmbH, Munich, Germany). However, in some experiments, the grids were placed directly into microplates at the beginning of the incubation cycle. This prevented damage to the delicate fimbrial and OMV structures, which would have occurred during scraping from the bottom of the microplates. After washing by dipping in dH_2_O, negative staining was conducted using 1% phosphotungstic acid diluted in ddH_2_O (Science Services GmbH) for a few seconds, and then the solution was soaked off and the grid was air-dried, which preserves the sample for examination. A Hitachi HT7800 TEM (Hitachi, Tokyo, Japan) operating at an acceleration voltage of 100 kV was used to perform the examination of the samples.

For SEM examination, first, 3% (vol/vol) glutaraldehyde (Agar Scientific, Rotherham, UK) in 0.1 M Sorensen’s phosphate buffer was used to fix the samples. Second, they were washed in phosphate buffer for 15 min and then dehydrated using an ascending ethanol series (30%, 50%, 70%, 90% and 100%). Each dehydration step lasted 10 min, and the last step was repeated three times. After being critical-point dried in liquid CO_2_ (Polaron critical-point dryer, GaLa Instrumente, Bad Schwalbach, Germany), the samples were sputter coated with a 10 nm layer of gold and palladium using a Sputter Coater EM SCD500 (Leica, Wetzlar, Germany). The analysis was performed using a SEM Quattro S (Thermo Fisher Scientific Inc.) with an acceleration voltage of 10 kV in a high-vacuum environment.

A subset of the SEM images from the multispecies experiments was manually colored based on morphology using Affinity Photo (version 1.10.8; Serif) image editing software to enhance the illustration of the interactions between species.

### 4.5. Vesicle Preparation

After prior incubation on TSBA, as described in Section 4.1., isolates of ATCC 49417 and OMI 1127 were anaerobically grown at 37 °C in 50 mL Falcon tubes (with 30 mL of Supplemented Brucella medium) for four days. Three biological replicates were utilized. The OMVs were isolated from the supernatant via filtration and differential centrifugation. The initial step in this process was centrifugation at 3000× *g* for 15 min to remove most of the cells. The resulting supernatant was filtered first through a 0.45 µm filter to remove residual bacteria and large cellular debris and then through a 0.2 µm membrane filter. Vesicles were then collected from this filtrate by ultracentrifugation, with the first step conducted at 10,000× *g* for 40 min, followed by further centrifugation at 100,000× *g* for 70 min. The resulting pellet was washed with PBS and then centrifuged again at 100,000× *g* for 70 min. The final ultracentrifugation was performed using a Beckman L-90K ultracentrifuge with a matching SW 32 Ti rotor. The OMV pellet was finally resuspended in PBS and stored at −80 °C. The presence of OMVs was confirmed by TEM. The pelleted OMVs were suspended in ddH_2_O and applied dropwise onto formvar carbon-coated nickel grids. The grids were air-dried, stained with 2% uranyl acetate and viewed under a Zeiss LEO 906E transmission electron microscope (Zeiss, Oberkochen, Germany). The size distribution and concentration of OMVs were determined by Nanoparticle Tracking Analysis (NTA), using a NanoSight NS300 (Malvern Instruments, Worcestershire, UK) equipped with a blue laser (488 nm, 70 mW). Analysis data of OMV preparations are provided by Appendix A. OMV samples were diluted 1:1000 in filtered PBS prior to NTA. Data analysis was performed with NTA 3.0 software (Malvern Instruments).

### 4.6. Proteomic Analysis of OMV Content

OMVs (three preparations from each strain) were prepared, as described in Section 4.5, and then lysed and proteolytically digested using the EasyPep Kit (A40006, Thermo Fisher Scientific, Dreieich, Germany) in accordance with the manufacturer’s instructions. Subsequently, the peptides were lyophilized and stored at −80 °C. Prior to mass spectrometry (MS) analysis, the samples were resuspended in 3% formic acid (FA)/1% acetonitrile (ACN) and loaded onto a nanoLC system (RSLCnano, Thermo Fisher Scientific Inc.), where they were first trapped on a trapping column (Acclaim PepMap100, C18, 5 µm, 100 Å, 300 µm i.d.  × 5 mm, Thermo Fisher Scientific Inc.) for 10 min. Thereafter, the peptides were separated using an analytical column at 45 °C (Aurora Ultimate 25 × 75 C18 UHPLC column, IonOpticks, Melbourne, Australia) with the following gradient: 0–10 min: 1% buffer B (buffer A: 0.1% FA; buffer B: 80% ACN, 0.1% FA), 10–12 min: 1–2% buffer B, 12–95 min: 2–25% buffer B, 95–120 min: 25–40% buffer B, 120–127 min: 40–99% buffer B, 127–132 min: 99% buffer B, 132–135 min: 99–2% buffer B, 135–150 min: 2% buffer B.

The data acquisition was performed on an Exploris 480 mass spectrometer (Thermo Fisher Scientific, Bremen, Germany) using a data-independent acquisition (DIA) method. This was carried out by using staggered acquisition windows (isolation window of 8 *m*/*z*, shift of 4 *m*/*z*). MS1 scans: 390–1010 or 394–1014 *m*/*z*. MS Settings: resolution 60k, normalized AGC target: 100%, max. injection time mode: 1; DIA settings: resolution 30k, normalized HCD collision energy: 30%, normalized AGC target: 1000%, max. injection time: 55 ms.

Analysis of the raw data was performed in Spectronaut (version 20.1, Biognosys AG, Schlieren, Switzerland) with all settings set as default except the identification settings, where all values were set to 0.01 [79]. MaxLFQ was used as the quantification method. The iBAQ values were utilized as the main output. The UniProt databases used were UP001179501 for *P. gingivalis* strain ATCC 49417 (OMI 1071) and UP001179540 for *P. gingivalis* strain OMI 1127 (84Pg1). In UP001179540, the error-prone entry tr|A0AAE9X6L4|A0AAE9X6L4_PORGN Fimbrillin was removed and replaced by the entry sp|Q93R80.2|FIMA6_PORGN as the valid HG1691-FimA sequence. Both databases were extended with a universal contaminants database [80] and are part of the *P. gingivalis* (strain ATCC BAA-308/W83) pan proteome in UniProt.

The resulting protein files were filtered for a minimum of two unique peptides (Spectronaut entry PG.NrOfStrippedSequencesIdentified), and a protein was required to be present in all three replicates (based on iBAQ values).

## 5. Conclusions

In this study, we gained insights into the architecture of long bundled fimbriae and OMVs, as well as the interaction between these two important *P. gingivalis* virulence factors, by analyzing a large number of electron microscope images. Our findings demonstrate that the bundling of long fimbriae occurs in all *P. gingivalis* strains that express them, though this may be obscured by strong capsule polysaccharide formation. Therefore, regarding our initial assumption, differences in the amino acid sequences of various FimA functional domains do not significantly affect bundling. Bundling also occurs between species (or even kingdoms), suggesting a role in adhesion to commensal bacteria and host cells. The tubular and chain-like protrusions of the outer membrane (OMEs) also appeared to interact with the two tested oral commensals. This, in turn, provides clues to their possible functions, including nutrient supply and/or communication. Future studies should attempt to elucidate their exact function in *P. gingivalis*.

Furthermore, our observations indicate that OMVs from the majority of strains can bind to (bundled) fimbriae forming FAVs. According to the proposed theory, accessory fimbrial proteins (FimCDE) could play a mediating role in this phenomenon. Their relatively high content in OMVs found in this study, combined with their proven contact-mediating properties, provide plausibility for this theory. It remains to be investigated whether OMVs are indeed transported along bundled fimbriae, mediated by FimCDE, as suggested by our images of the FAVs and the proteome analysis. In addition, FimCDE may also play a pivotal role in the formation and stabilization of bundled fimbriae. Subsequent studies must determine the exact localization, composition and number of complexes of the three proteins within the bundled fimbriae and investigate their function in greater depth.

## Figures and Tables

**Figure 1 ijms-27-00383-f001:**
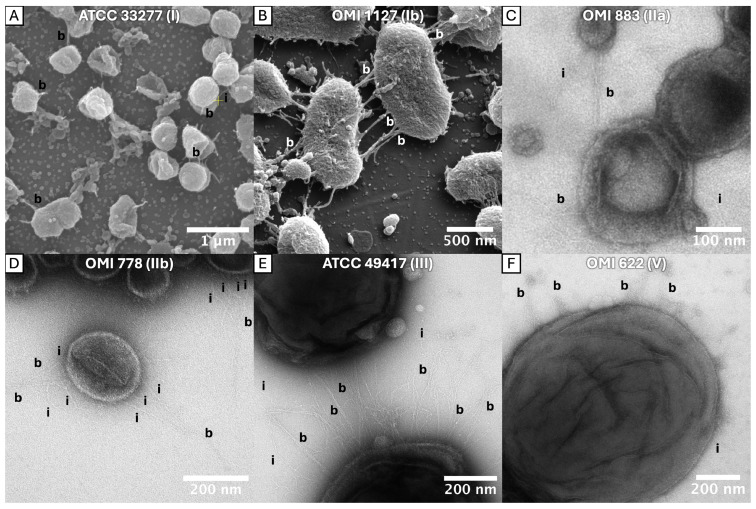
SEM (**A**,**B**) and TEM (**C**–**F**) micrographs of *P. gingivalis* with long bundled fimbriae. The strain designations and *fimA* type are indicated within each panel. Individual and bundled fimbriae are marked with “i” and “b”, respectively. The diameter of the bundles ranges from approximately 10 to 50 nm, depending on the numbers of fimbriae involved. In contrast, individual fimbria had a maximum diameter of 5 nm. Bundled fimbriae protruded from the cell surface, or in (**D**), from the vesicle surface, into the medium. In (**B**,**C**,**E**), the fimbriae of the opposing cells have become entangled.

**Figure 2 ijms-27-00383-f002:**
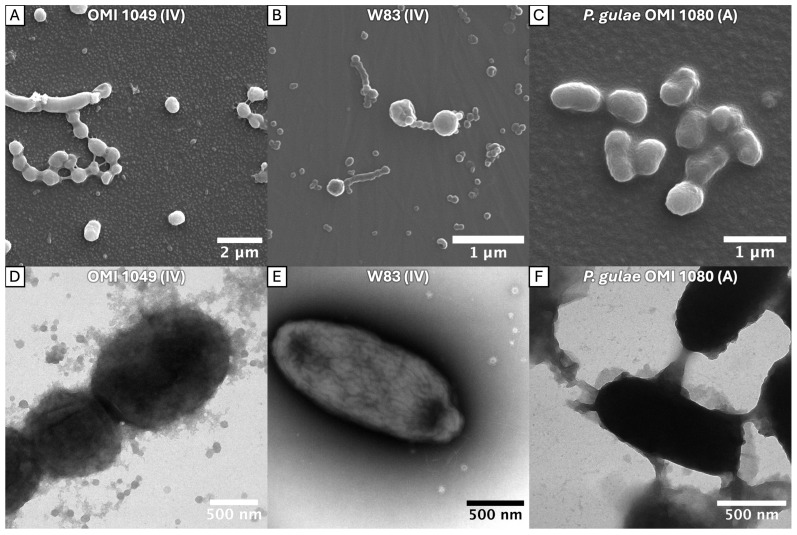
*P. gingivalis* FimA type IV strains (AJW5 = OMI 1049 and afimbriated W83 = OMI 629) and *P. gulae* (OMI 1080, fimbrial type A) in SEM (**A**–**C**) and TEM (**D**–**F**) micrographs. W83 and the *P. gulae* strain formed a strong capsule and many vesicles but showed no (bundled) fimbriae. In slimy strain OMI 1049, bundled fimbriae are masked but can still be assumed, as depicted in image (**A**).

**Figure 3 ijms-27-00383-f003:**
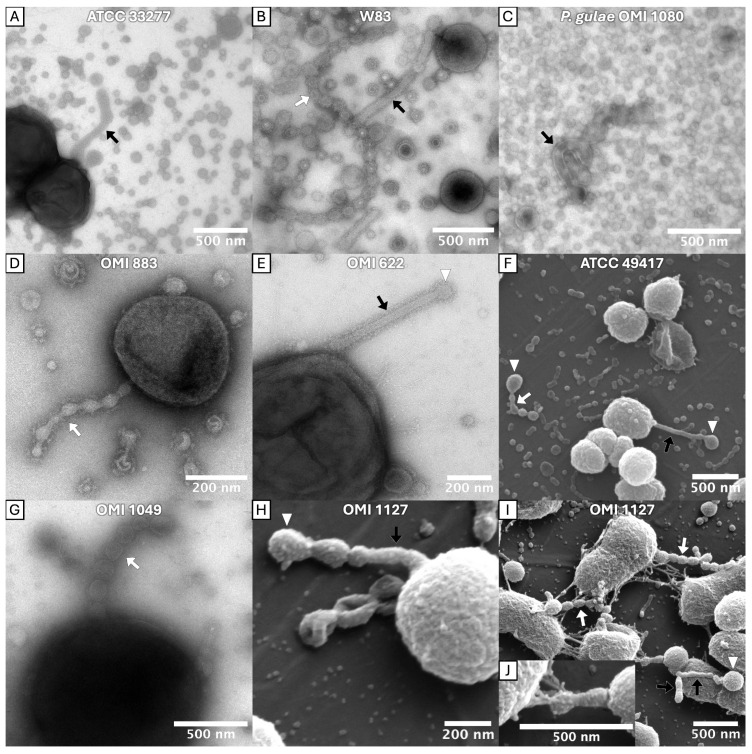
TEM (**A**–**E**,**G**) and SEM (**F**,**H**–**J**) micrographs of tubular and chain-like OMEs. *P. gingivalis* formed tubes (indicated with ➡) that protruded from the cell surface and chains of OMVs that appeared to be interconnected (indicated with ⇨). The transitions between the two forms appeared fluid. Probably due to the preparation process, some of these structures were detached from the cell. Vesicles with a larger diameter were frequently observed at the end of these structures (indicated with ▷). Tubes adhered to bundled fimbriae (**J**). In TEM images, a prominent EDSL was visible on the OMVs and on the chain- and tube-like OMEs. Strain designations are indicated within the panels.

**Figure 4 ijms-27-00383-f004:**
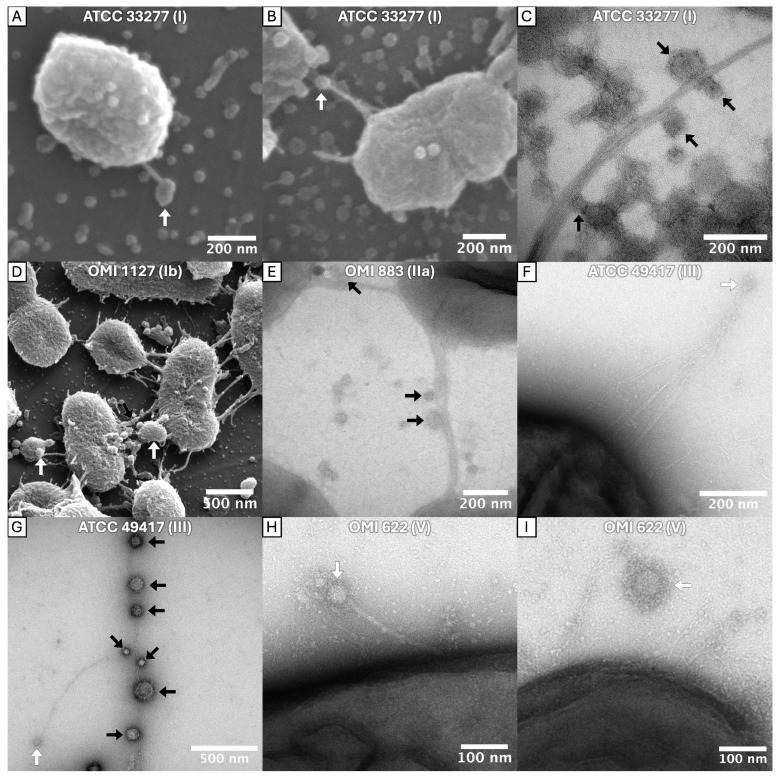
SEM (**A**,**B**,**D**) and TEM (**C**,**E**–**I**) micrographs showing OMVs associated with *P. gingivalis* long fimbriae. OMVs attached to both long single and bundled fimbriae and commonly appeared at the distal tips of fimbriae (indicated with ⇨) or at multiple points along their shaft (indicated with ➡). The TEM micrographs displayed a prominent EDSL on the OMVs. The strain designations and *fimA* type are indicated within each panel.

**Figure 5 ijms-27-00383-f005:**
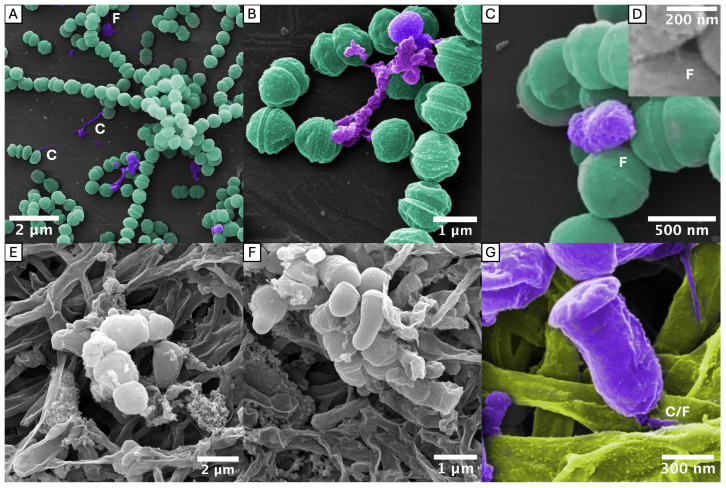
*P. gingivalis* co-cultured with *S. oralis* (top row) and *F. nucleatum* (bottom row) documented by SEM micrographs. To facilitate differentiation, in some images, *P. gingivalis* was highlighted in purple, *S. oralis* in green and *F. nucleatum* in yellow. *P. gingivalis* cells were found individually and in small colonies between the predominant *S. oralis* cells (**A**). (**B**,**C**) illustrate specific details from (**A**). In (**B**), two *P. gingivalis* cells were observed to adhere to *S. oralis*, subsequently forming a network with the *S. oralis* cells. This network likely contained (bundled) fimbriae, OMVs and matrix components. In section (**C**) ((**D**) = enlarged detail), a *P. gingivalis* cell appears to have been attached to the *S. oralis* cell at its lower edge by a short robust bundle of fimbriae. In the *F. nucleatum* co-culture, slightly larger *P. gingivalis* cell clusters were identified (**E**,**F**). A prevalent finding was the presence of an altered polar cell morphology of *P. gingivalis* cells (**F**,**G**). (**G**) shows this in detail. One side of the cell exhibits a kind of cap, while the other side appears to attach to *F. nucleatum* with a bundle of fimbriae. Bacteria were incubated anaerobically in co-culture for 5–7 days. The coloration was performed using image processing software (Affinity Photo, version 1.10.8, Serif). C = chain(s) of OMVs, F = (bundled) fimbriae.

**Figure 6 ijms-27-00383-f006:**
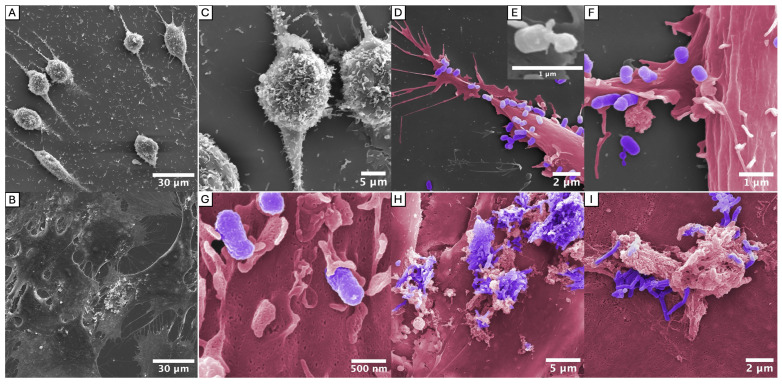
HGFs co-incubated with *P. gingivalis* ATCC 49417 (**A**,**C**–**G**) and OMI 1127 (**H**,**I**) in comparison to the control of pure HGF culture (**B**) documented by SEM micrographs. HGFs in the co-culture with ATCC 49417 rounded up (**A**,**C**) compared to the pure culture (**B**), indicative of stress due to anaerobic conditions and/or bacterial toxins. ATCC 49417 exhibited extensive colonization of the HGFs (**C**). The bacterial cells adhered more readily to the HGFs than to the glass slides (**C**,**D**). At high magnification, direct interactions between *P. gingivalis* and the fibroblasts were visible (**E**,**G**). Bundled fimbriae were visible between *P. gingivalis* and HGFs ((**E**) = enlarged section of (**D**)). Similar to the co-culture with *F. nucleatum*, stalked adhesion with partially asymmetrical deformation of the bacteria could be observed (**F**). In contrast, OMI 1127 was concentrated in proximity to fibroblast products, most likely collagen (as its substrate), and was hardly identifiable on the remaining surfaces (**H**,**I**). The SEM images were subsequently colored using image processing software (Affinity Photo, version 1.10.8, Serif) to facilitate differentiation between *P. gingivalis* (purple) and HGFs (pink). The samples were incubated anaerobically for 24 h in 75% DMEM (with 10% FBS) and 25% Brucella medium.

**Figure 7 ijms-27-00383-f007:**
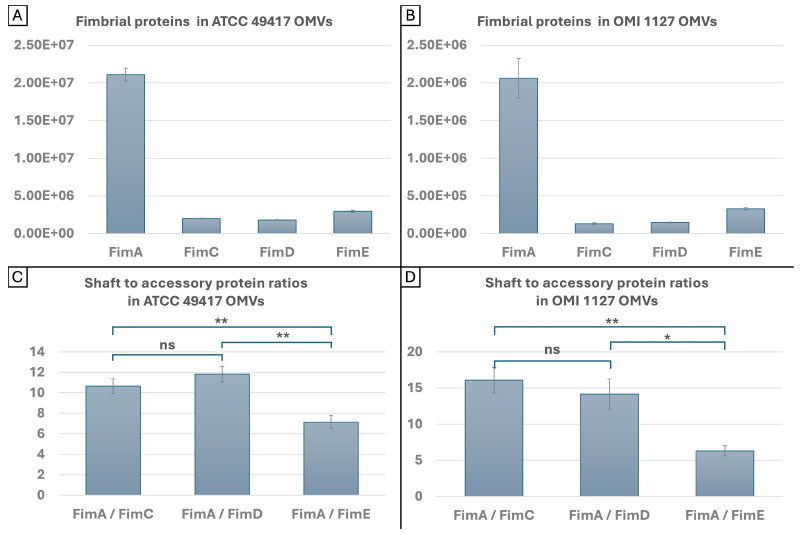
Quantities (**A**,**B**) and ratios (**C**,**D**) of shaft and accessory proteins of major fimbriae in OMV preparations of two *P. gingivalis* strains. (**A**,**B**): Graphical representation of the determined MVs (n = 3) listed in Appendix A. Of the four proteins, FimA is the most abundant, followed by FimE, then FimC and FimD. The quantities are not absolute and cannot be compared between preparations. (**C**,**D**): Graphical representation of the determined MRs of the shaft to each accessory protein (n = 3). The ratio of FimA to FimE was significantly smaller than those of FimA to FimC and FimD in both strains. In contrast, the ratios of FimA to FimC and FimD were similar. The iBAQ values were determined using DIA proteomics software (Spectronaut). SDs are displayed within the bars. Statistical analysis was performed using unpaired *t*-tests with *p* < 0.05 (ns: non-significant, *: *p* < 0.05, **: *p* < 0.01).

**Figure 8 ijms-27-00383-f008:**
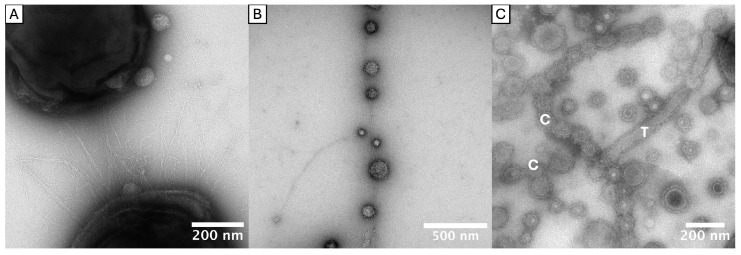
Overview of the observed structures. TEM micrographs of *P. gingivalis*. (**A**) Bundled fimbriae, (**B**) FAVs, (**C**) tube- and chain-like OMVs. T = tube, C = chain.

**Figure 9 ijms-27-00383-f009:**
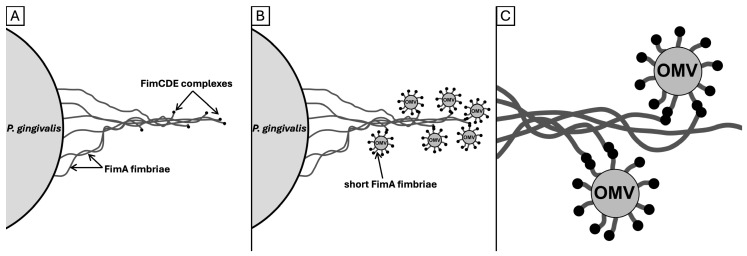
New hypothesis about the spatial positioning and aggregatory function of the accessory/tip FimCDE complex expressed on OMV (shorter) and cellular (longer, bundled) fimbriae. (**A**) Assuming that fimbriae are of equal length, it can be deduced that the tip proteins FimCDE become located at different positions alongside the bundle since the fimbriae of a bundle are anchored at different distances from the “central” fimbriae on the membrane. (**B**) FimCDE as the tips on short FimA fimbriae complexes of OMVs interact with the FimCDE of long bundled FimA fimbriae. Thus, the OMVs are distributed at several positions along the bundled fimbriae, as shown in Figure 4G and Figure 8B. (**C**) OMVs may bind multiple fimbriae concurrently via FimCDE-FimCDE interactions, thereby contributing to the stabilization or formation of the bundle. The figure was created using image processing software (Affinity Photo, version 1.10.8, Serif).

**Table 1 ijms-27-00383-t001:** Bacterial strains used in this study and type of fimbriae in the case of *Porphyromonas*.

Strain Designation	Original Code	Id Name Species	Fimbriae Type ^1^
OMI 1080	G 251	*P. gingivalis* animal biotype/*P. gulae*	A
OMI 1132	ATCC 33277	*P. gingivalis*	I
OMI 1127	84Pg1	*P. gingivalis*	Ib
OMI 883	AC 27	*P. gingivalis*	IIa
OMI 778	AC 38	*P. gingivalis*	IIb
OMI 1071	ATCC 49417, RB22D	*P. gingivalis*	III
OMI 629	W83	*P. gingivalis*	IV
OMI 1049	AJW5 (VAG5)	*P. gingivalis*	IV
OMI 622	Fr. 025/15-1	*P. gingivalis*	V
OMI 581	ATCC 35037	*S. oralis*	-
OMI 1040	ATCC 25586	*F. nucleatum*	-

^1^ based on variations in the nucleotide sequences of *fimA* and amino acid sequences of FimA.

## Data Availability

The data presented in this study are available on request from the first or senior author (J.L., G.C.). The mass spectrometry proteomics data have been deposited with the ProteomeXchange Consortium via the PRIDE [81] partner repository with the dataset identifier PXD070506.

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
