# Peer review of "Porphyromonas gingivalis Bundled Fimbriae Interact with Outer Membrane Vesicles, Commensals and Fibroblasts"

_ijms, 2025, doi:10.3390/ijms27010383_

Round 1
Reviewer 1 Report
Comments and Suggestions for Authors
The authors of this study have attempted to visualize the formation of bundled fimbriae in different P. gingivalis FimA types and to monitor their interactions with outer membrane vesicles, commensal bacteria as well as human gingival fibroblasts. They have also analyzed protein content of OMVs from two different strains and shed light on possible functions of additional fimbirae proteins (FimCDE). Bundling of long fimbriae has been supported by SEM/TEM pictures for the first time in literature only recently, and previous authors have mostly examined Fim A and B, while FimCDE and their functions are still under-investigated, which this study is attempting to correct. Therefore, the subject of this paper is novel and a significant contribution to the field. The manuscript as a whole is well written and has appropriate references. The Introductions provides enough information and addresses previous contributions as well as emphasizing the gap of knowledge, especially regarding the role of under-investigated FimCDE, which is one of the subjects of this study. The methodology is appropriate, well executed and explained, and the figures well presented, with detailed legends. The Results are clearly presented and the conclusions, especially those regarding the possible functions of FimCDE, supported by them. Also, the authors have stated the necessity of further investigations regarding the localization and exact mechanisms of FimCDE function.
Since this study further elucidates the roles of long fimbriae of P. gingivalis, especially the possible role of FimCDE in mediating the transport of outer membrane vesicles, as well as their involvement in formation and stabilization of bundled fimbriae, I do believe that this paper will be of great interest to readers and should be published in the present form.
Author Response
Comments 1: The authors of this study have attempted to visualize the formation of bundled fimbriae in different P. gingivalis FimA types and to monitor their interactions with outer membrane vesicles, commensal bacteria as well as human gingival fibroblasts. They have also analyzed protein content of OMVs from two different strains and shed light on possible functions of additional fimbirae proteins (FimCDE). Bundling of long fimbriae has been supported by SEM/TEM pictures for the first time in literature only recently, and previous authors have mostly examined Fim A and B, while FimCDE and their functions are still under-investigated, which this study is attempting to correct. Therefore, the subject of this paper is novel and a significant contribution to the field. The manuscript as a whole is well written and has appropriate references. The Introductions provides enough information and addresses previous contributions as well as emphasizing the gap of knowledge, especially regarding the role of under-investigated FimCDE, which is one of the subjects of this study. The methodology is appropriate, well executed and explained, and the figures well presented, with detailed legends. The Results are clearly presented and the conclusions, especially those regarding the possible functions of FimCDE, supported by them. Also, the authors have stated the necessity of further investigations regarding the localization and exact mechanisms of FimCDE function.
Since this study further elucidates the roles of long fimbriae of P. gingivalis, especially the possible role of FimCDE in mediating the transport of outer membrane vesicles, as well as their involvement in formation and stabilization of bundled fimbriae, I do believe that this paper will be of great interest to readers and should be published in the present form.
Response 1: Thank you for your supportive summary of our study. We appreciate the time you spent carefully reading and reviewing it.
Reviewer 2 Report
Comments and Suggestions for Authors
The manuscript describes a comprehensive electron microscopic study of Porphyromonas gingivalis fimbriae (type V pili) in diverse FimA types. The study shows that fimbrial bundling or braiding is a general property. Also, the bacterium expresses membranous surface extensions and secreted vesicles. Both the fimbriae and the membranous structures participate in homotypic and heterotypic cell aggregation. The manuscript also discusses positioning and putative role for the fimbrial accessory proteins FimC, FimD, and FimE. The work is novel and comprehensive, and the micrographs are of high quality.
An important omission is that Fig. 8 does not show a control HGF cells incubated anaerobically in the absence of bacteria. This control is essential for interpretation of bacterial effects on the cells.
- 561-562: A procedural issue is potential lack of fixation for TEMs of OMVs and membrane extensions. Were cells fixed before washing in water? If not, then the visualized structures might be artefacts of cell lysis, and additional controls are needed..
Textual issues by line number:
57-58: It is not clear from the context whether Mfa1 corresponds to a separate gene or just a fimbrial phenotype.
58: “A third adhesin…” Please designate first and second adhesins
77: I suggest “… the polymerization process, because long fimbriae…”
79: I suggest “…FimD, but neither protein was mounted…”
82: I suggest “…all three mutants had diminished…”
89-90: Can the text be specific about the mechanism of immune subversion?
92: Spelling for “deaminases”?
91-94: sentence not clear. What determines preference for TLR1 or TLR6?
94-95: This sentence states that OMVs themselves are invasive. That seems incorrect.
101 &102: Define “brick wall” and “cell-cell knotting.”
126: Cell connections cannot appear to be slimy in EM. They may be amorphous.
126: What is the observational basis for the statement that bundled fimbriae are responsible?
133: P. gulae is a non-human pathogen and/or commensal. It is not an “animal analogue.”
Figure 1; l. 147: Mark individual and bundled fimbriae, especially in panel D.
162: I suggest “….electron micrographs of negatively stained preparations…”
165: The structures marked appear to be straight and kinked, rather than “serpentine,” a term that implies a sinuous structure.
166-167: the electron dense layer should be marked on vesicles and cells or referenced.
185-186: “...with the exception of the latter, also in P. gulae…” The meaning is not clear.
205: define “EDSL.”
298 and elsewhere: define and use iBAC or IBAC consistently
314: “…the data…”
352: The braids are especially visible in Figure 4C.
372-373: Relevant data may exist from the study in Kim et al., Front Oral Health 2021 doi: 10.3389/froh.2021.701659. Supp. Figure 5
458 & 480: “anchor” is the correct word
467 ff and Fig. 9: Some bundles in the figures are 500-1000 nm long. This observation impIies that some fimbriae must be that long. Therefore, the FimCDE ratios might also result from lateral binding to FimA. Please discuss.
638: “…can be obscured…” This statement is not justified by the evidence presented. I suggest “…may be obscured…”
Author Response
Comments 1: The manuscript describes a comprehensive electron microscopic study of Porphyromonas gingivalis fimbriae (type V pili) in diverse FimA types. The study shows that fimbrial bundling or braiding is a general property. Also, the bacterium expresses membranous surface extensions and secreted vesicles. Both the fimbriae and the membranous structures participate in homotypic and heterotypic cell aggregation. The manuscript also discusses positioning and putative role for the fimbrial accessory proteins FimC, FimD, and FimE. The work is novel and comprehensive, and the micrographs are of high quality.
Response 1: Thank you very much for the overall support of our study.
Comments 2: An important omission is that Fig. 8 does not show a control HGF cells incubated anaerobically in the absence of bacteria. This control is essential for interpretation of bacterial effects on the cells.
Response 2: Thank you very much for bringing up this important issue. HGF reactions are provided with Figure 6 (not 8 to be exact). Actually we are providing the control with Figure 6B. However, to make it more clear we have modified the Figure 6 caption:
Figure 6. HGFs co-incubated with P. gingivalis ATCC 49417 (A, C-G) and OMI 1127 (H, I) in comparison to the control of pure HGF culture (B) documented by SEM micrographs. HGFs in the co-culture with ATCC 49417 rounded up (A, C) compared to pure culture (B) indicative for stress due to anaerobic conditions and/or bacterial toxins.
Comments 3: 561-562: A procedural issue is potential lack of fixation for TEMs of OMVs and membrane extensions. Were cells fixed before washing in water? If not, then the visualized structures might be artefacts of cell lysis, and additional controls are needed..
Response 3: In our experience the fixation with aldehydes negatively affects the negative staining method by creating a too strong dark staining which covers small structures. Therefore, we avoided fixation on purpose. Once applied to the grid the samples air dry within 5 minutes and are then dried and fixed. We do not expect cell lysis within this short time. To make this point clearer, however, we added text to our M&M section 4.4.: After washing by dipping in dH2O, negative staining was conducted using 1% phosphotungstic acid diluted in ddH2O (Science Services GmbH) for a few seconds, then the solution was soaked off and the grid was air dried, what preserves the sample for examination.
Textual issues by line number:
Comments 4: 57-58: It is not clear from the context whether Mfa1 corresponds to a separate gene or just a fimbrial phenotype.
Response 4: Thank you very much. The genes are encoded by different gene clusters; we have modified the sentences accordingly:
- gingivalis fimbriae are among the most potent pathogenic factors for host colonization [15] and evasion of the innate immune system [16]. P. gingivalis expresses various types of fimbriae by different gene clusters: First, the FimA fimbriae, also termed ‘major’ or ‘long fimbriae’, and second, the Mfa1 fimbriae, also termed ‘minor’ or ‘short fimbriae’
Comments 5: 58: “A third adhesin…” Please designate first and second adhesins
Response 5: Thank you very much, we have modified the sentences accordingly (see above and in text).
Comments 6: 77: I suggest “… the polymerization process, because long fimbriae…”
Response 6: We have changed accordingly:
It has been established that they are non-essential for the polymerization process, because long fimbriae are formed in knock-out mutants [32,33].
Comments 7: 79: I suggest “…FimD, but neither protein was mounted…”
Response 7: We have changed accordingly:
In more detail, a fimE insertion mutant still produced FimC and FimD, but neither protein was mounted on the lengthening fimbriae.
Comments 8: 82: I suggest “…all three mutants had diminished…”
Response 8: We have changed accordingly:
Purified fimbriae from all three mutants had diminished binding activities to glyceraldehyde-3-phosphate dehydrogenase (GAPDH)
Comments 9: 89-90: Can the text be specific about the mechanism of immune subversion?
Response: We have changed accordingly:
Furthermore, the mutant failed to exploit CXC-chemokine receptor 4 (CXCR4) to suppress TLR2 signaling, resulting in impaired immune subversion in vivo. Purified FimC and FimD (but not FimE) were shown to interact with CXCR4 [35].
Response 9: We have checked: peptidyl arginine deiminases (PPAD) is the correct spelling (as in the title of reference cited).
Comments 10: 91-94: sentence not clear. What determines preference for TLR1 or TLR6?
Response 10: The reviewer is right; we have deleted the corresponding phrase as literature about this topic is indeed unclear.
Comments 11: 94-95: This sentence states that OMVs themselves are invasive. That seems incorrect.
Response 11: We have checked: Indeed; the vesicles actually enter HGFs according to Ref. Mantri et al. The exact text in the reference says “[…] a comparison of the expression levels of fimC, fimD, and fimE in P. gingivalis strains and their respective vesicle invasive capabilities implicates the minor components of long fimbriae as having a role in vesicle entry into HGFs.”
Comments 12: 101 &102: Define “brick wall” and “cell-cell knotting.”
Response 12: We now provide definitions of these terms. Thank you.
The study of Meyer et al. [25] revealed a wide spectrum of fimbrial superstructures for the first time, including bundling, cell-cell knotting (P. gingivalis cells connected via bundled fimbriae) and brick-wall formation (regular, brickwork-like arrangements of cells) in strain ATCC 33277 [25].
Comments 13: 126: Cell connections cannot appear to be slimy in EM. They may be amorphous. Same line: What is the observational basis for the statement that bundled fimbriae are responsible?
Response 13: Thank you for this critical comment. We provide more explanations:
…the cell-connections were found less frequently and appeared more amorphous, though they could still be assumed to be bundled fimbriae given their elongated, narrow, hourglass-like shape (Fig. 2 A). Compared to the others, this strain's cultures appeared slimy on agar, suggesting increased production of capsule polysaccharides, and such slime could explain the amorphous appearance.
Comments 14: 133: P. gulae is a non-human pathogen and/or commensal. It is not an “animal analogue.”
Response 14: The reviewer is right; we have deleted this phrase.
Comments 15: Figure 1; l. 147: Mark individual and bundled fimbriae, especially in panel D.
Response 15: We have included many examples of individual (“i”) and bundled (“b”) fimbriae into our Figure 1 graphs. Also, we have added the sentence “Individual and bundled fimbriae are marked with “i” and “b”, respectively” to our Figure 1 caption.
Comments 16: 162: I suggest “….electron micrographs of negatively stained preparations…”
Response 16: We would like to keep the original version, also to avoid doubling of “preparation(s)” in two consecutive sentences.
Comments 17: 165: The structures marked appear to be straight and kinked, rather than “serpentine,” a term that implies a sinuous structure.
Response 17: Thank you for this thoughtful comment. We have changed the sentence into:
These tubes exhibited either kinked (Fig. 3A), winding (Fig. 3C) or straight (Fig. 3E) conformations.
Comments 18: 166-167: the electron dense layer (EDSL) should be marked on vesicles and cells or referenced. Line 2025: define “EDSL”
Response 18: We decided not to mark individual EDSL protein complexes on the vesicles because there are so many of them, and we think they are apparent in the images at many sites. However, we added the following text to better define/demonstrate EDSL:
A prominent electron-dense surface layer (EDSL: an outer membrane-located, immunostimulant, granular protein-rich layer, which constitutes a major portion of P. gingivalis OMV’s volume [44,45]) was apparent on the tubular extensions and most clearly visible in Fig. 3B, D, and E.
Comments 19: 185-186: “...with the exception of the latter, also in P. gulae…” The meaning is not clear.
Response 19: Thank you. You are right. We have re-written the sentence to make it clearer.
Taken together, tubular OMEs, OMV chains, and FAVs were found across diverse P. gingivalis strains and, with exception of the FAVs, also in P. gulae.
Comments 20: 298 and elsewhere: define and use iBAC or IBAC consistently
Response 20: We have added the following sentence:
In order to estimate absolute protein quantities the Intensity-based absolute quantification (iBAQ) algorithm (see [51], supplementary methods) was used. The iBAQ values were determined by analysis of the mass spectrometry raw data with Spectronaut and the calculated mean values (MVs) are displayed in Suppl. Tables S1 and S2.
Comments 21: 314: “…the data…”
Response 21: Thanks, corrected; you have a good eye.
Comments 22: 352: The braids are especially visible in Figure 4C.
Response 22: Thanks, corrected; the reviewer is right, 4C.
Comments 23: 372-373: Relevant data may exist from the study in Kim et al., Front Oral Health 2021 doi: 10.3389/froh.2021.701659. Supp. Figure 5
Response 23: We could not find information on the relationship between fimbriae, pili -on one side- and capsule expression -on the other side- in the recommended study/supp. fig. This study further supports our statement, backed by other studies mentioned earlier in the text, that the presence of a capsule reduces P. gingivalis’ surface adhesion.
Comments 24: 458 & 480: “anchor” is the correct word
Response 24: Thanks, corrected; you have a good eye.
Comments 25: 467 ff and Fig. 9: Some bundles in the figures are 500-1000 nm long. This observation impIies that some fimbriae must be that long. Therefore, the FimCDE ratios might also result from lateral binding to FimA. Please discuss.
Response 25: Thank you for this comment supporting our hypothesis. Indeed we discuss the possibility of more than one FimCDE complex per fimbriae and a few lines further down of the lines you are mentioning. “An alternative explanation for the occurrence of OMVs in the course of the fimbrial bundles and at least to some extent for the altered, reduced ratios of FimA to the accessory proteins could be, that FimCDE complexes were not only located at the tip of the fimbriae, as it has been postulated so far [26,33], but also down- (anchor-) stream. […] To provide substantiating evidence, the development and utilization of fluorescently labeled monoclonal antibodies targeting FimCDE would be necessary to test the hypothesis. Ultimately, a synergy between both phenomena (firstly, different positions where FimCDE complexes come to lie, and secondly, multiple FimCDE complexes along the long fimbriae) would also be plausible.“
Comments 26: 638: “…can be obscured…” This statement is not justified by the evidence presented. I suggest “…may be obscured…”
Response 26: Thanks, improved according to your suggestion.
We appreciate the time you spent carefully reading and reviewing it.
Reviewer 3 Report
Comments and Suggestions for Authors
This is a very well presented and rigorous study on a topical subject in oral microbiology. I have no suggestions for further content improvement. There is a minor spelling correction to be made: in lines 458 and 480, "anker" should be "anchor."
Author Response
Comments 1: This is a very well presented and rigorous study on a topical subject in oral microbiology. I have no suggestions for further content improvement. There is a minor spelling correction to be made: in lines 458 and 480, "anker" should be "anchor."
Response 1: Thank you for the nice supportive summary of our study. We corrected the typo; you have a good eye.
Round 2
Reviewer 2 Report
Comments and Suggestions for Authors
All comments have been addressed.